# Improving Consistency in Retrieval Augmented Systems with Group Similarity Reward

## Abstract

RAG systems are increasingly deployed in high-stakes domains where users expect outputs to be consistent across semantically equivalent queries. However, existing systems often exhibit significant inconsistencies due to variability in both the retriever and generator (LLM), undermining trust and reliability. In this work, we focus on *information consistency*—the requirement that outputs convey the same core content and information across semantically equivalent inputs. We introduce a principled evaluation framework that decomposes RAG consistency into retriever-level, generator-level, and end-to-end components, helping identify inconsistency sources. To improve consistency, we propose **P**araphrased **S**et Group Relative Policy Optimization (PS-GRPO), an RL approach that leverages multiple rollouts across paraphrased set to assign *group similarity rewards*. We leverage PS-GRPO to achieve Information **Con**sistent **RAG** (Con-RAG), training the generator to produce consistent outputs across paraphrased queries and remain robust to retrieval-induced variability. Because exact reward computation over paraphrase sets is computationally expensive, we also introduce a scalable approximation method that retains effectiveness while enabling efficient, large-scale training. Empirical evaluations across short-form, multi-hop, and long-form QA benchmarks demonstrate that Con-RAG significantly improves both consistency and accuracy over strong baselines, even in the absence of explicit ground-truth supervision. Our work provides practical solutions for evaluating and building reliable RAG systems for safety-critical deployments.

## 1 Introduction

LLMs are increasingly used in open-domain applications where users expect them to behave predictably, producing consistent outputs for semantically equivalent or paraphrased inputs. However, they frequently generate divergent responses to such variations, raising concerns about their reliability (Novikova et al., 2025; Elazar et al., 2021; Raj et al., 2025). RAG systems are particularly prone to such inconsistencies (Perçin et al., 2025). These architectures combine a retriever and a generator: the retriever selects top-$k$ documents from a large corpus based on the query, and the generator synthesizes a response conditioned on those documents (Gao et al., 2023). Semantically similar queries can lead to different retrieved document sets or rankings, resulting in divergent outputs (Perçin et al., 2025). Recent work by Weller et al. (2025) also highlights a theoretical bottleneck in embedding-based retrieval, showing that the expressivity of top-$k$ retrieval is fundamentally limited—underscoring the need for systems that are robust to retrieval inconsistencies. Furthermore, even when the evidence is fixed, the generator may still produce inconsistent responses due to the non-deterministic nature and phrasing sensitivity of LLMs (Razavi et al., 2025).

This inconsistency is particularly problematic in high-stakes domains such as healthcare, finance, or legal settings, where RAG systems are commonly deployed (Kim et al., 2025). Inconsistent outputs can erode trust, introduce liability risks, or even mislead users (Kim et al., 2025; Novikova et al., 2025). For instance, a customer service RAG assistant may offer different instructions for "*How do I close my savings account?*" and "*What steps should I take to shut down my savings account?*" despite these queries being semantically equivalent (Razavi et al., 2025).

In this work, we focus on *information consistency*—the requirement that outputs convey the same core content and information across paraphrased inputs (see motivational Figure 1). This contrasts with

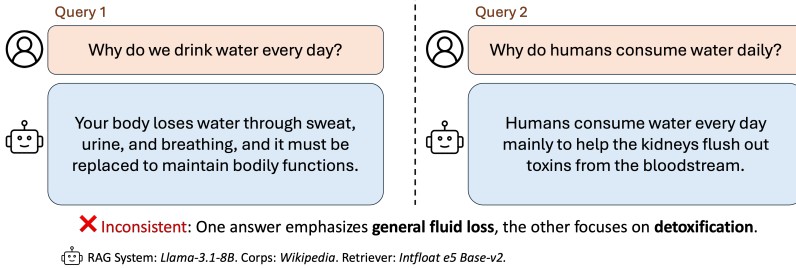

Figure 1: **Motivational Example.** Two semantically equivalent queries lead to different outputs from a RAG system, despite both responses being factually correct. Such variation may be acceptable in many applications, but in certain high-stakes domains (e.g., healthcare, finance, legal) information consistency across semantically equivalent inputs may be required to ensure reliability, user trust, and compliance.

*lexical consistency*, which emphasizes word-level or structural similarity. While lexical consistency is easier to measure, it can penalize legitimate variation (e.g., use of synonyms or stylistic changes) and is insufficient in evaluating factual agreement. Crucially, the relationship between consistency and accuracy varies across QA tasks. In short-form QA, where answers are typically concise and factual, improving consistency often correlates with higher accuracy, models that are more consistent tend to be more correct. In contrast, for long-form QA tasks, where multiple valid answers may exist, consistency and accuracy become orthogonal dimensions: a model can be accurate yet inconsistent, or vice versa. Hence, in open-ended tasks, enforcing information consistency becomes a key desideratum alongside answer quality.

Given the practical importance of consistent outputs, we aim to address the following question: *How can we measure & improve the information consistency of RAG system outputs across semantically equivalent inputs, without compromising factual accuracy?* To tackle this, we introduce a new evaluation framework that decomposes consistency into retriever-level and generator-level components, and propose a reinforcement learning approach to optimize for consistency using group similarity rewards. Our contributions can be summarized as follows:

- **A Framework for Measuring Consistency in RAG Systems.** We present a principled framework to evaluate consistency in RAG systems by disentangling three components: retriever consistency (Jaccard overlap of documents), generator consistency (LLM outputs given fixed context), and end-to-end consistency. We instantiate this using lexical and LLM-Judge based similarity metrics, offering insights into where and how inconsistencies emerge (see Section 2.1).

- **Con-RAG: Improving Consistency via Paraphrased Set GRPO.** To enhance consistency across semantically equivalent queries, we propose **P**araphrased **S**et Group Relative Policy Optimization (PS-GRPO), an RL approach that leverages multiple rollouts across a set of paraphrased inputs to assign *group similarity rewards*. This forms the core of our Information **Con**sistent **RAG** (Con-RAG) framework (see Figure 3). Due to complexity of computing the rewards, we introduce a relaxed approximation by subsampling paraphrases and rollouts, reducing the number of comparisons from quadratic to linear in the number of paraphrases. This allows us to train Con-RAG efficiently on large datasets while preserving reward fidelity (see Section 2.2).

- **Empirical Evaluation.** We conduct an extensive evaluation of Con-RAG across five QA benchmarks: short-form, multi-hop, and long-form QA tasks (see Figure 2). Our results show that Con-RAG significantly improves both end-to-end and generator consistency over a wide range of baselines, without degrading accuracy. In long-form QA tasks, Con-RAG improves both consistency and LLM-judged factual accuracy despite being trained in the absence of explicit ground-truth supervision. We evaluate using Llama3.1 and Qwen2.5 model families.

**Related Work.** *Consistency in Language Models:* Consistency has emerged as a key concern for safety and reliability in high-stakes LLM deployment (Kim et al., 2025; Novikova et al., 2025). Prior work has introduced various notions of consistency. Logical consistency refers to the ability of the model to make decisions without logical contradiction (Jang et al., 2022; Li et al., 2019; Asai & Hajishirzi, 2020; Mitchell et al., 2022). Factual consistency, often discussed as faithfulness or

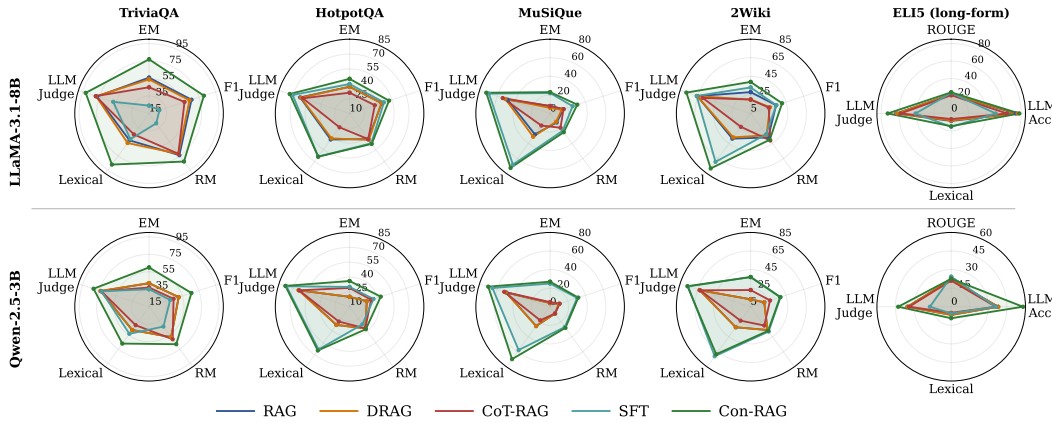

Figure 2: **Comparison between Con-RAG and baselines across accuracy and consistency dimensions on LLaMA-3.1-8B and Qwen-2.5-3B.** Each plot summarizes performance on a single dataset using accuracy measures (Exact Match, token F1, Relaxed Match) and end-to-end information consistency (measured lexically and via LLM-judge). Con-RAG consistently outperforms prior methods across models, achieving both higher factual accuracy and more consistent responses across paraphrased inputs (see Table 2 for full numerical results).

hallucination, considers whether model outputs contradict the source content (Jang et al., 2022; Wang et al., 2020; Maynez et al., 2020; Tam et al., 2022). Self-consistency evaluates whether similar inputs yield stable explanations (Parcalabescu & Frank, 2023). Nonlogical forms of consistency, such as moral consistency, assess coherence of values across contexts (Bonagiri et al., 2024; Arvanitis & Kalliris, 2020). Prediction consistency to model multiplicity (Hamman et al., 2025; Gomez et al., 2024). Closest to our work is semantic consistency, which measures output stability under semantically equivalent inputs like paraphrases. This has been evaluated using datasets like ParaRel (Elazar et al., 2021) and metrics such as BERTScore, entailment scores, and LLM judges (Raj et al., 2022; Rabinovich et al., 2023; Kuhn et al., 2023). Approaches to improve semantic consistenc include custom losses (Elazar et al., 2021), knowledge distillation from consistent teachers (Raj et al., 2025), and synthetic data supervision (Zhao et al., 2024b). We refer to a recent survey exploring current landscape, challenges, and future directions in consistency in LLMs (Novikova et al., 2025).

*Consistency in RAG Systems:* RAG improves factual accuracy by conditioning outputs on retrieved evidence (Guu et al., 2020; Karpukhin et al., 2020; Lewis et al., 2020). However, it introduces new sources of inconsistency due to retriever sensitivity and generator (LLM) variability. Despite growing use in high-stakes applications, information consistency in RAG remains underexplored, with the exception a few notable studies addressing robustness in retrieval or prompt-level variation (Hsia et al.; Zhang et al., 2025; Hu et al., 2024; Perçin et al., 2025). Our work aims to evaluate and improve information consistency in RAG, leveraging an RL-based optimization with paraphrase group similarity rewards. Our approach builds on recent advances in RL for LLMs (Kaufmann et al., 2024), particularly GRPO (Shao et al., 2024), which trains on verifiable reward assignment across outputs. Our framework improves information consistency across semantically equivalent inputs without relying on strong supervision or ground-truth labels, unlike prior methods.

## 2 MAIN CONTRIBUTIONS

In this section, we first define a framework to measure consistency in RAG systems by isolating retriever, generator, and end-to-end contributions (see Section 2.1), then introduce our Con-RAG method to improve consistency via group similarity rewards and its relaxation (see Section 2.2).

### 2.1 MEASURING CONSISTENCY IN RAG SYSTEMS

We consider a RAG system composed of a retriever $R$ and a generator (LLM). Given a user query $q$, the system first retrieves a set of top-$k$ documents from a corpus $\mathcal{D}$, and then generates an

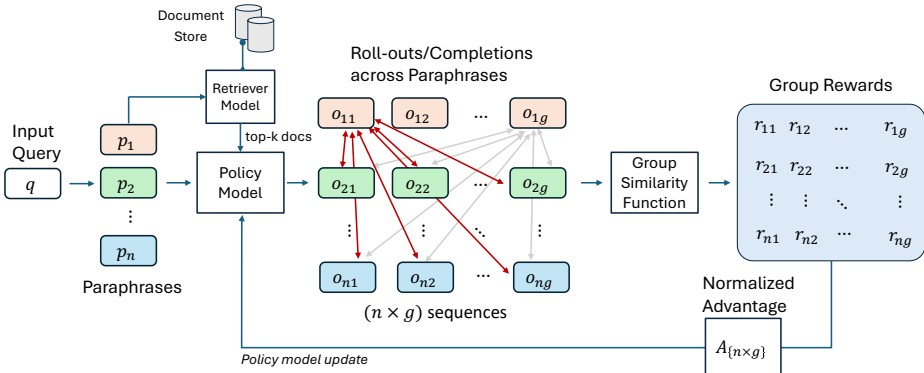

Figure 3: Overview of PS-GRPO and Information **Con**sistent **RAG** (Con-RAG) framework. A canonical query $q$ is expanded into a set of paraphrases $\{p_1, \ldots, p_n\}$, each of which is passed through the policy LLM to generate $g$ sampled rollouts. For every rollout $o_{ij}$, we compute a paraphrase group similarity reward $r_{ij}$ by averaging its similarity with outputs from other paraphrases of the same query (this produces an $n \times g$ reward matrix). Normalized advantages are then computed within each paraphrase set, and the policy model is updated.

output $y = \text{LLM}(q, R(q))$ conditioned on these documents: $R(q) = \{d_1, \ldots, d_k\} \subset \mathcal{D}$. Let $q_0$ be a canonical input query, and let $\mathcal{P}(q_0) = \{p_1, p_2, \ldots, p_n\}$ denote a set of paraphrased or semantically equivalent inputs. Our goal is to assess the *output consistency* of the RAG system across this paraphrased set.

**Retriever Consistency.** Let $R(p_i)$ denote the set of documents retrieved for paraphrase $p_i$. We define retriever-level consistency as the average similarity between the document sets retrieved for all pairs of paraphrases. We use *Jaccard similarity* (Gower & Legendre, 1986), which measures the ratio of the intersection to the union of two sets. This metric directly captures the overlap between retrieved evidence sets while normalizing for their total size. The overall retriever consistency is then the average across all unique paraphrase pairs: $\mathcal{C}_{\text{ret}}(q_0) = \frac{2}{n(n-1)} \sum_{i,j} \frac{|R(p_i) \cap R(p_j)|}{|R(p_i) \cup R(p_j)|}$.

**End-to-End RAG Consistency.** Let $y_i = \text{LLM}(p_i, R(p_i))$ denote the output of the RAG system for paraphrase $p_i$. End-to-end consistency measures alignment across outputs when the entire pipeline is allowed to vary, each paraphrase $p_i$ is passed to the retriever, which may return a different document set $R(p_i)$, and the generator then conditions on this evidence to produce $y_i$. Formally, we compute pairwise similarity across all outputs: $\mathcal{C}_{\text{gen}}(q_0) = \frac{1}{n(n-1)} \sum_{i \neq j} \text{sim}(y_i, y_j)$. This captures the overall stability of the RAG system under paraphrased inputs, reflecting the combined variability introduced by both retrieval and generation. The similarity function $\text{sim}(\cdot, \cdot)$ can be instantiated using various metrics, including lexical similarity (e.g., BLEU, ROUGE), embedding-based similarity (e.g., BERTScore), entailment-based scores from NLI models, or LLM-based judgments using a strong language model to assess consistency or contradiction between $y_i$ and $y_j$.

**Generator (LLM) Consistency.** To isolate the generator's contribution, we can fix the retrieved documents across all paraphrases and measure similarity among the outputs, i.e., $y_i^{\text{fixed}} = \text{LLM}(p_i, R(q_0))$, and compute consistency over $\{y_1^{\text{fixed}}, \ldots, y_n^{\text{fixed}}\}$. This captures how consistently the LLM alone responds to semantically equivalent inputs when conditioned on identical evidence. Conceptually, this is closely related to prior work on consistency in standalone LLMs, where the focus is on ensuring paraphrase-invariant outputs under identical or similar prompts (Elazar et al., 2021; Raj et al., 2025; Novikova et al., 2025; Razavi et al., 2025).

## 2.2 IMPROVING CONSISTENCY VIA PARAPHRASED SET GRPO

Given a RAG system comprise a retriever $R$ and generator (LLM). A canonical query $q_0$ with paraphrases $\mathcal{P}(q_0) = \{p_1, \ldots, p_n\}$, our goal is to maximize output consistency without degrading factual accuracy. We propose Paraphrased Set GRPO, an RL algorithm that leverages GRPO's multiple rollouts across paraphrased inputs to assign group-level similarity rewards. Our objective is to directly optimize the generator so that outputs across semantically equivalent inputs are consistent.

**Group Relative Policy Optimization.** GRPO is RL optimization algorithm that estimates advantage through group-normalized rewards rather than using a critic model (Shao et al., 2024). For a given query $q$, GRPO samples a group of $g$ rollouts $\{o_1, \ldots, o_n\}$ from the current policy $o_i \sim \pi_\theta(\cdot \mid q)$, assigns each a verifiable scalar reward $r_i = \text{Reward}(o_i|q)$, and computes group-normalized advantages $\hat{A}_i = (r_i - \mu_q)/\sigma_q$, where $\mu_q$ and $\sigma_q$ are the mean and standard deviation of rewards within the $g$ rollouts. Let $y_{i,1:|o_i|}$ denote tokens of response $o_i$ and $\rho_{i,t} = \frac{\pi_\theta(y_{i,t}|p,y_{i,<t})}{\pi_{\theta_{old}}(y_{i,t}|p,y_{i,<t})}$. The policy is then optimized by maximizing the objective using these group-relative advantages, with an optional KL penalty to penalize deviation from the reference policy:

$$\mathcal{L}_{\text{GRPO}}(\theta) = \frac{1}{g} \sum_{i=1}^{g} \sum_{t=1}^{|o_i|} \min\Big(\rho_{i,t}\hat{A}_i, \text{clip}(\rho_{i,t}, 1-\epsilon, 1+\epsilon)\hat{A}_i\Big) - \beta \, \mathbb{D}_{\text{KL}}\big(\pi_\theta(\cdot \mid q) \,\big\|\, \pi_{\text{ref}}(\cdot \mid q)\big) \quad (1)$$

**Group Similarity Reward.** PS-GRPO introduces a group-level objective that promotes consistent generation across semantically equivalent queries. It leverages the unique property of GRPO which generates *extensive rollouts per query*. We aggregate all rollouts from all paraphrases into a single group and compute similarity-based rewards *across the paraphrase dimension*, so each output is rewarded according to its similarity with outputs generated for the other paraphrases of the same canonical query. For each canonical query $q_0$ with paraphrases $\mathcal{P}(q_0) = \{p_1, \ldots, p_n\}$, the policy LLM $\pi_\theta$ generates $g$ rollouts per paraphrase: $o_{ij} \sim \pi_\theta(\cdot \mid p_i, R(p_i)), i \in \{1, \ldots, n\}, j \in \{1, \ldots, g\}$. Collect these into an $n \times g$ matrix $\{o_{ij}\}$ (total $n \times g$ rollouts). We assign each rollout $o_{ij}$ a paraphrase group similarity reward by averaging its similarity to all rollouts generated for the *other* paraphrases (also see Figure 3 for illustration):

$$r_{ij} = \frac{1}{(n-1)g} \sum_{\substack{u=1 \\ u \neq i}}^{n} \sum_{m=1}^{g} \text{sim}\big(o_{ij}, o_{um}\big), \quad (2)$$

where $\text{sim}(\cdot, \cdot)$ is the agreement function. In practice, we instantiate sim using the BLEU metric, motivated by recent findings that BLEU serves as a surprisingly strong proxy for reward models in aligning LLMs with human preferences (Chang et al., 2025). As further validated in our ablation study (see Table 4), BLEU consistently outperformed alternative similarity metrics while remaining computationally efficient. Group-normalized advantages are then computed across each paraphrased rollout: $\hat{A}_{ij} = (r_{ij} - \mu_i)/\sigma_i$, with $\mu_i, \sigma_i$ the mean and standard deviation of rewards for rollouts for $p_i$. The policy is optimized with the standard GRPO clipped objective using $\hat{A}_{ij}$ and (optionally) a KL penalty to a reference policy with weight $\beta$. If ground-truth answers are available (e.g., in short-form QA tasks), we extend the reward to improve consistency and accuracy. Specifically, for each rollout we define a combined weighted reward:

$$r_{ij}^{\text{final}} = \alpha \, r_{ij}^{\text{cons}} + \gamma \, \text{Acc}(o_{ij}, y^\star), \quad (3)$$

where $r_{ij}^{\text{cons}}$ is the group similarity reward, $y^\star$ is the ground-truth answer, and $\text{Acc}(\cdot, \cdot)$ is measured using token F1 score. Importantly, our method does not require ground truths to improve consistency: the accuracy reward term can be omitted, as demonstrated in our long-form QA experiments (see Section 3), where questions are open-ended and no single ground-truth answer exists.

**Efficient Computation of Group Similarity Rewards for Scalable Training.** Computing paraphrase group similarity rewards can be expensive, especially in a training environment where rewards must be computed at every gradient step. This overhead can significantly slow down training. For each rollout $o_{ij}$, computing its reward requires comparing against all $(n-1)g$ rollouts from the other paraphrases. At the query level, with $n$ paraphrases and $g$ rollouts each, the naive total cost is $ng \times (n-1)g = n(n-1)g^2$ similarity computations. For example, with $n = 5$ and $g = 6$ amounts to 720 similarity comparisons for a single query. Exploiting symmetry (a similarity between $o_{ij}$ and $o_{um}$ need not be recomputed twice) reduces this to $\frac{1}{2}n(n-1)g^2$, but the cost still scales quadratically with $n$ and $g$. To make training feasible, we introduce a *relaxed group similarity reward*. Instead of averaging over all cross-paraphrase comparisons, for each rollout $o_{ij}$ we subsample $\kappa$ paraphrases $K \subset \{1, \ldots, n\} \setminus \{i\}$ and $s$ rollouts per chosen paraphrase, and approximate: $\tilde{r}_{ij} = \frac{1}{\kappa s} \sum_{u \in K} \sum_{m \in S_k} \text{sim}(o_{ij}, o_{um})$, which is an unbiased estimator under uniform sampling. This reduces the per-query cost from $O(n(n-1)g^2)$ to $O(ng\kappa s)$, if $\kappa \ll n-1$ and $s \ll g$. In practice, this approximation preserves the training signal for cross-paraphrase consistency while keeping the reward computation tractable.

Table 1: **Disentangling sources of inconsistency in RAG systems (`LLaMA-3.1-8B`)**. Retriever consistency is low across datasets, suggesting that paraphrased queries often retrieve non-overlapping documents. This introduces context variability that is reflected in the end-to-end consistency scores. Fixing retrieval improves consistency, but variation remains, revealing the generator's sensitivity to input phrasing even with identical evidence. We present accuracy values in Table 5 (also see Table 7 for Qwen-2.5-3B).

| Dataset | End-to-End Consistency | | Generator (LLM) Consistency | | Retriever Consistency |
|---------|-------|-----------|-------|-----------|----------------|
|         | Lexical | LLM-Judge | Lexical | LLM-Judge | Jaccard Overlap |
| TriviaQA | 53.0 | 77.8 | 67.3 | 88.5 | 32.5 |
| HotpotQA | 42.5 | 62.5 | 53.7 | 71.9 | 46.0 |
| 2Wiki | 38.5 | 65.5 | 48.4 | 76.4 | 52.4 |
| MuSiQue | 27.9 | 48.2 | 44.4 | 69.7 | 36.6 |
| Eli5 | 8.56 | 62.8 | 15.1 | 74.2 | 27.1 |

## 3 EXPERIMENTS

In this section, we describe our experimental setup to evaluate the effectiveness of Con-RAG across diverse QA tasks, outlining our datasets, paraphrase generation, consistency metrics, training details, and comparisons with competitive baselines.

**Datasets.** We evaluate our approach across three types of question answering (QA) tasks: Short-form QA tasks: TriviaQA (Joshi et al., 2017) and HotpotQA (Yang et al., 2018), both requiring concise fact-based answers. Multi-hop QA tasks: 2WikiMultiHopQA (Ho et al., 2020) and MuSiQue (Trivedi et al., 2022), which involve reasoning over multiple pieces of evidence. Long-form QA task: ELI5 (Fan et al., 2019), where answers are open-ended and typically span multiple sentences. None of these datasets provide paraphrased versions of the input questions. To evaluate consistency, we synthetically generate paraphrases for each query.

**Generating paraphrased and semantically equivalent queries.** For each query $q_0$, we use LLaMA 3.1 70B to generate $n$ paraphrases $\mathcal{P}(q_0) = \{p_1, \ldots, p_n\}$. To ensure answerability, we provide the ground truth answer as part of the prompt and instruct the model to generate paraphrases that preserve the exact meaning such that each paraphrase can be answered in the same way. This allows us to simulate semantically equivalent inputs without altering the expected outputs (see prompt in Appendix A.2).

**Setup.** Our RAG system consists of a LLaMA 3.1 8B and Qwen 2.5 3B model serving as the generator, and a dense retriever built on top of the intfloat/e5-base-v2 embedding model (Wang et al., 2022). We use KILT Wikipedia snapshot (Petroni et al., 2020) as our document corpus, where each article is segmented into chunks of 512 tokens before embedding. All embeddings are indexed using FAISS for efficient retrieval. At inference time, the retriever selects the top-$k = 5$ documents per query, which are then appended to the prompt for generation. To isolate effects from sampling inconsistencies, we use deterministic decoding throughout all experiments.

**Evaluating Consistency in RAG Systems.** We evaluate performance along two dimensions: accuracy and consistency. For short-form and multi-hop QA datasets, accuracy is measured using: (i) Exact Match (EM), (ii) token F1 score, and (iii) Relaxed Match (RM), which considers an answer correct if the ground truth answer appears anywhere in the output. For long-form QA (e.g., ELI5), where answers are open-ended and may be phrased in diverse ways, EM/F1/RM are too restrictive. Instead, we evaluate accuracy using: (i) ROUGE, to capture content overlap with reference answers, and (ii) LLM-judge accuracy, where a strong model (LLaMA 3.3 70B) assesses whether the generated answer is factually correct. Consistency is evaluated at three levels (disentangling contributions from the retriever and generator): (i) End-to-end consistency, where each paraphrase retrieves its own documents and we compute agreement between outputs (via BLEU for lexical consistency and an LLM judge for information consistency—see Appendix A.3 for prompts); (ii) Generator consistency, where retrieval is fixed across paraphrases and agreements cross outputs are measured; (iii) Retriever consistency, defined as the average Jaccard overlap between retrieved document sets across paraphrases (see Section 2.1). We use paraphrase size $n = 5$ for evaluations.

We summarize consistency results across the datasets in Table 1. We observe that the retriever consistency is relatively low across the datasets, indicating that paraphrases often retrieve non-

Table 2: **Comparison between Con-RAG vs. Baselines (Short-form QA Tasks) (LLaMA 3.1 8B).** Lexical consistency measured via BLEU score while and information consistency measured using an LLM-judge. Con-RAG is trained with a group-similarity reward plus an accuracy reward (no KL), and consistently yields higher end-to-end and generator-only consistency while also improving accuracy over original queries (see radar plot illustration in Figure 2). Refer to Table 9 for results on Qwen-2.5-3B model.

| Dataset | Method | Accuracy (%) | | | End-to-End Consistency (%) | | Generator (LLM) Consistency (%) | |
|---|---|---|---|---|---|---|---|---|
| | | EM | F1 | RM | Lexical | Inform. | Lexical | Inform. |
| TriviaQA | RAG | 56.0 | 66.1 | 74.0 | 53.0 | 77.8 | 67.3 | 88.5 |
| | DRAG | 54.0 | 63.7 | 72.0 | 56.8 | 78.7 | 68.2 | 88.2 |
| | CoT-RAG | 45.0 | 57.7 | 72.0 | 44.6 | 79.2 | 57.7 | 85.0 |
| | SFT | 24.0 | 27.5 | 29.0 | 51.3 | 58.2 | 77.8 | 81.2 |
| | **Con-RAG** | **77.0** | **81.0** | **83.0** | **87.3** | **91.3** | **91.2** | **93.0** |
| HotpotQA | RAG | 37.0 | 44.1 | 42.0 | 42.5 | 62.5 | 53.7 | 71.9 |
| | DRAG | 37.0 | 43.8 | 43.0 | 41.1 | 61.6 | 50.5 | 73.1 |
| | CoT-RAG | 31.0 | 36.8 | 42.0 | 27.3 | 59.6 | 36.1 | 68.9 |
| | SFT | 39.7 | 46.5 | 47.2 | **63.9** | 70.5 | 72.2 | 78.5 |
| | **Con-RAG** | **45.0** | **51.9** | **48.0** | **63.9** | **73.6** | **80.9** | **88.2** |
| MuSiQue | RAG | 8.0 | 15.3 | 12.0 | 27.9 | 48.2 | 44.4 | 69.7 |
| | DRAG | 6.0 | 13.1 | 11.0 | 31.0 | 50.7 | 42.9 | 70.0 |
| | CoT-RAG | 8.0 | 15.2 | 19.0 | 16.1 | 53.7 | 29.2 | 67.7 |
| | SFT | 22.0 | 25.5 | 23.0 | 68.1 | 69.3 | 77.8 | 79.8 |
| | **Con-RAG** | **23.0** | **30.8** | **25.0** | **72.5** | **72.3** | **91.4** | **92.7** |
| 2Wiki | RAG | 28.0 | 33.9 | 37.0 | 38.5 | 65.5 | 48.4 | 76.4 |
| | DRAG | 20.0 | 26.9 | 34.0 | 36.8 | 65.5 | 49.3 | 76.1 |
| | CoT-RAG | 20.0 | 25.5 | 41.0 | 22.8 | 59.3 | 29.9 | 67.8 |
| | SFT | 33.0 | 34.0 | 33.0 | 69.4 | 66.2 | 84.4 | 83.3 |
| | **Con-RAG** | **39.0** | **40.6** | **40.0** | **78.2** | **77.8** | **94.1** | **95.5** |

Table 3: **Comparison between Con-RAG vs. Baselines (Long-form QA Task).** Con-RAG is trained using only the group-similarity reward with a small KL regularizer (no accuracy supervision). Despite no ground-truth, it achieves the best end-to-end and generator consistency and also improves answer quality over baselines, whereas SFT on reference answers underperforms in this open-ended setting.

| Dataset | Method | Accuracy (%) | | End-to-End Consistency (%) | | Generator (LLM) Consistency (%) | |
|---|---|---|---|---|---|---|---|
| | | ROUGE | LLM-Acc | Lexical | Inform. | Lexical | Inform. |
| ELI5 | RAG | 21.9 | 74.0 | 8.6 | 62.8 | 15.1 | 74.2 |
| | DRAG | 22.0 | 76.0 | 8.0 | 62.2 | 15.0 | 72.5 |
| | CoT-RAG | 20.9 | 64.0 | 6.4 | 57.8 | 10.3 | 71.0 |
| | SFT | 23.5 | 51.0 | **15.3** | 40.8 | 16.6 | 41.7 |
| | **Con-RAG** | **24.2** | **78.0** | 14.6 | **72.7** | **21.7** | **80.8** |

overlapping sets of documents, a key source of downstream inconsistency. This is reflected in the end-to-end consistency scores, which shows that these small changes in query phrasing can result in different answers, due to shifts in both retrieved context and model generation. To isolate the generator's contribution, we also evaluate generator consistency under fixed retrieval (i.e., same documents across paraphrases). While consistency scores improve, substantial variability still remain, showing that even with identical evidence, the generator (LLM) exhibits sensitivity to input phrasing.

We report accuracy for original queries, paraphrased queries, and paraphrased queries with fixed documents in Table 8. Across these settings, accuracy remains relatively stable, with only minor fluctuations, suggesting that paraphrasing and retrieval shifts have limited impact on final answer correctness on average.

**Con-RAG Training Setup.** We train Con-RAG with BLEU as similarity function for computing group similarity rewards. For short-form and multi-hop QA tasks, we use unigram BLEU (ngram=1)

Table 4: **Effect of Reward Similarity Metric on Con-RAG (`ELI5- Qwen2.5-3B`).** We vary the similarity function used in the group reward to study its impact on information consistency. Lower-order BLEU emphasizes word choice and local fluency, aligning better with the goal of preserving core information across paraphrases. In contrast, higher-order BLEU and Exact Match enforce stricter surface-level or sentence-level overlap, which can penalize valid rephrasings. BLEU-2 yields the best consistency and accuracy, indicating that rewarding semantic adequacy is better aligned with information consistency.

| Reward Metric | Accuracy (%) | | End-to-End Cons. (%) | | Generator Cons. (%) | |
|---|---|---|---|---|---|---|
| | ROUGE | LLM-Acc | Lexical | LLM-Judge | Lexical | LLM-Judge |
| BLEU-1 | **22.6** | 54.0 | 6.9 | 38.2 | 14.8 | **69.8** |
| BLEU-2 | 22.5 | **58.0** | **9.2** | **42.0** | **17.8** | 67.5 |
| BLEU-3 | 22.4 | 49.0 | 6.7 | 36.3 | 14.8 | 66.0 |
| BLEU-4 | 22.2 | 50.0 | 6.4 | 36.2 | 14.2 | 66.5 |
| ROUGE-L | 22.1 | 46.0 | 6.1 | 35.2 | 13.6 | 65.2 |
| Exact Match | 22.1 | 49.0 | 6.6 | 37.7 | 14.4 | 66.2 |

and bigram BLEU (ngram=2) for long-form QA tasks to account for more contextual similarity across longer answers. For short-form QA tasks, where ground-truth answers are available, we augment the similarity reward with an accuracy reward based on token F1 score, which we found to be more stable than other accuracy metrics. The final reward is computed using a weighted sum as defined in Eq. 3, with equal weights $(\alpha, \gamma = 1)$ for both consistency and accuracy. We set the KL regularization coefficient $\beta = 0.0$ for these tasks, following recent findings (Hu et al., 2025) suggesting that GRPO performs effectively without explicit KL penalties. In contrast, for long-form QA (ELI5), where questions are open-ended and multiple valid answers may exist, we exclude the accuracy reward and optimize solely for consistency using the group similarity reward. To prevent reward hacking in the absence of ground-truth supervision, we apply a small KL penalty with $\beta = 0.05$ to regularize the policy against a reference model.

We use $n = 6$ paraphrases per canonical query and $g = 4$ rollouts per paraphrase. To make training scalable, we apply the relaxed approximation described in Section 2.2 to estimate group similarity rewards. Specifically, we subsample $\kappa = 3$ paraphrases and $s = 1$ rollout per selected paraphrase when computing similarity, which significantly reduces the number of comparisons with minimal impact on reward quality. We perform full model fine-tuning using the AdamW optimizer with a learning rate of `1e-6`. All training is conducted on LLaMA 3.1 8B and Qwen2.5-3B.

**Baselines.** We compare Con-RAG against diverse baselines representative of current RAG systems: (i) **RAG**: A standard RAG setup where the top-k retrieved documents are appended to the prompt and passed directly to the generator for answer prediction. (ii) **DRAG** (Demonstrated RAG) (Yue et al., 2024): An inference-time scaling method that leverages few-shot demonstrations to improve performance. (iii) **CoT-RAG** (Chain-of-Thought RAG) (Zhao et al., 2024a): Extends standard RAG by prompting the generator to produce intermediate reasoning steps before outputting a final answer, improving multi-hop and compositional question answering. (iv) **SFT** (Supervised Fine-Tuning) (Chung et al., 2024): We fine-tune the generator on paraphrased queries paired with their ground-truths. For long-form QA, where answers are free-form, we fine-tune on the available reference responses. (v) **Con-RAG** (ours): Our proposed method that leverages group similarity rewards to improve consistency (see Section 2.2). All baselines are evaluated using the same retriever, generator, and document corpus to ensure fair and consistent comparison.

**Results and Analysis.** We present our results across short-form and long-form QA tasks in Figure 2 and Tables 2. To show that consistency improvements do not come at the cost of answer quality, we report accuracy metrics on the original queries, avoiding generic but consistent outputs. Our results demonstrate the following key observations:

*Con-RAG improves both consistency and accuracy in short-form QA.* Across all short-form and multi-hop datasets, Con-RAG achieves significant gains in both end-to-end and generator-only consistency. For instance, on TriviaQA, end-to-end consistency (lexical/information) improves from $53.0/77.8$ (RAG) to $87.3/91.3$, while generator consistency reaches $91.2/93.0$. Notably, these improvements are not achieved at the expense of accuracy. Con-RAG also achieves the highest EM, F1, and RM scores across all datasets. This indicates that optimizing consistency can also enhance model robustness, likely due to the implicit data augmentation effect of training across paraphrase groups.

Table 5: **Effect of Accuracy Reward Variant on Con-RAG (`TriviaQA - Qwen2.5-3B`).** We compare consistency-only training, accuracy-only training, and joint training with consistency plus various accuracy metrics. The best performance is achieved when combining consistency with the token F1 reward, which yields the highest accuracy and consistency values.

| Reward Variant | $\alpha$ | $\gamma$ | Accuracy (%) | | | End-to-End Cons. (%) | | Generator Cons. (%) | |
|---|---|---|---|---|---|---|---|---|---|
| | | | EM | F1 | RM | Lexical | LLM-Judge | Lexical | LLM-Judge |
| Consistency only | 1.0 | 0.0 | 51.5 | 53.2 | 59.0 | 59.9 | 79.0 | 78.7 | 88.0 |
| Accuracy only (F1) | 0.0 | 1.0 | 54.0 | 56.0 | 60.4 | 52.0 | 75.0 | 62.0 | 84.1 |
| Consistency + EM | 1.0 | 1.0 | 56.2 | 63.5 | 65.0 | 61.5 | 80.2 | 76.0 | 88.4 |
| Consistency + RM | 1.0 | 1.0 | 57.0 | 64.0 | 66.0 | 62.3 | 80.5 | 77.0 | 88.5 |
| Consistency + F1 | 1.0 | 1.0 | **60.0** | **66.0** | **68.0** | **67.1** | **81.8** | **80.5** | **89.5** |

Other baselines DRAG and CoT-RAG provide only modest consistency improvements and fail to match Con-RAG across metrics.

*In Long-form QA, Con-RAG also boosts accuracy without ground-truth supervision.* Results on ELI5 (see Table 3) are particularly interesting: even though Con-RAG is trained without any explicit ground truth (or accuracy signal), it improves both consistency and accuracy over all baselines. Compared to RAG, Con-RAG increases lexical and information consistency while also achieving higher ROUGE and LLM-judged accuracy. In contrast, SFT trained on reference answers performs poorly on ELI5, especially in terms of LLM-judge accuracy, highlighting the limitations of rigid supervision in open-ended QA, where many valid responses exist. This underscores the strength of Con-RAG in open-ended tasks, which does not rely on a single reference output.

**Ablation Studies.** To analyze design choices in Con-RAG, we run focused ablations on a lighter generator, Qwen2.5 3B for fast, controlled sweeps. *1.) Varying similarity function used in the group reward.* We replace BLEU in the group similarity reward with alternative choices and measure resulting consistency/accuracy. We consider: BLEU-$n$ ($n \in \{1, 2, 3, 4\}$), ROUGE-L, Exact Match (results are summarized in Table 4). *2.) Varying short-form accuracy reward metrics.* On short-form QA, we study the effect of different reward signals on accuracy and consistency by conducting ablations with: (i) consistency term only training, (ii) accuracy term only training, and (iii) joint training with consistency plus accuracy. For the accuracy component, we compare token F1 (ours), EM, and RM (see Table 5). *3.) Effect of LLM decoding temperature on consistency and accuracy.* We evaluate how inference-time stochasticity impacts consistency and accuracy by sweeping temperature values $T \in \{0.0, 0.5, 1.0, 2.0\}$ during decoding (see results in Table 11).

**Discussion.** While Con-RAG achieves strong improvements in both generator and end-to-end consistency, several important directions remain as next steps. (1) *Beyond Lexical Rewards for Information Consistency:* In this work, we use lexical similarity metrics (e.g., BLEU) as a proxy to enforce information consistency. While effective, such metrics emphasize surface-level alignment and penalize variations in wording, even when the underlying information remains unchanged. In practice, we may allow use of synonyms or outputs expressed differently, as long as they convey the same core content. A key next step is to search for a signal that would directly optimize for information-level consistency without enforcing lexical similarity between outputs. LLM as a judge seems promising, however, such a signal introduces a tension between weak vs. strong supervision (Burns et al., 2023). Ideally, we seek lightweight, automatic signals that can still guide the model toward consistent output (leveraging entailment-based rewards, BERTScore, etc.). (2) *Joint Retriever and Generator Optimization:* Con-RAG substantially improves generator consistency, yet end-to-end consistency still lags behind, mainly due to variation in retrieved documents across paraphrased queries. This inconsistency in retrieval results in different contexts being provided to the generator. To address this, a promising next step is to jointly optimize the retriever and generator. By rewarding the retriever to return similar documents for semantically equivalent queries, and simultaneously training the generator for consistency, the system can learn to retrieve relevant evidence that best helps answer the question accurately, potentially further improving both consistency and accuracy (Lewis et al., 2020). By introducing a principled way to measure RAG consistency and a scalable method to improve it, we move toward more reliable, trustworthy, and user-aligned RAG systems.

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

# A APPENDIX

## A.1 BLEU: AN $n$-GRAM BASED EVALUATION METRIC

The BLEU (Bilingual Evaluation Understudy ([Papineni et al.](), [2002])) score is a standard metric for assessing the quality of machine translation. It quantifies the degree of overlap between a system-generated translation and one or more human reference translations. The score relies on modified $n$-gram precision ($n \in \{1, 2, 3, 4\}$), together with a brevity penalty (BP) that discourages excessively short outputs:

$$\text{BLEU} = \text{BP} \cdot \exp\left(\sum_{n=1}^{N} w_n \log p_n\right), \quad \text{BP} = \begin{cases} 1 & \text{if } c > r, \\ \exp\left(1 - \frac{r}{c}\right) & \text{if } c \leq r, \end{cases}$$

where $p_n$ denotes the modified $n$-gram precision, $w_n$ are the associated weights, $c$ is the length of the candidate translation, and $r$ is the length of the closest reference. When multiple references are given, BLEU counts $n$-gram matches against all references and uses the maximum match count for each $n$-gram. Each $n$-gram level in BLEU captures progressively deeper aspects of linguistic quality. Unigrams ($n=1$) assess word choice or adequacy, indicating whether the candidate includes the correct content words. Bigrams ($n=2$) begin to reflect local fluency by capturing short-range word ordering. Trigrams ($n=3$) provide signals about phrase-level coherence, identifying whether multi-word chunks align with natural phrasing. 4-grams ($n=4$) enforce sentence-level fluency by requiring longer, contiguous sequences to match the reference.

## A.2 GENERATING PARAPHRASED AND SEMANTICALLY EQUIVALENT QUERIES.

For each query $q_0$, we use LLaMA 3.1 70B to generate $n$ paraphrases $\mathcal{P}(q_0) = \{p_1, \ldots, p_n\}$. To ensure answerability, we provide the ground truth answer as part of the prompt and instruct the model to generate paraphrases that preserve the exact meaning such that each paraphrase can be answered in the same way. This allows us to simulate semantically equivalent inputs without altering the expected outputs. See prompt used for short-form and long form QA tasks below:

---

**Paraphrasing – Short-form QA**

You are given an input sentence. Your task is to generate n diverse paraphrases of this sentence. You can paraphrase by using synonyms, changing sentence structure, or rephrasing in any other way, but each paraphrase should preserve the original meaning. Each paraphrase you create must be answerable by the exact same answer provided below.
Format your output as follows:
`<paraphrase1>` paraphrased sentence 1 `</paraphrase1>`
`<paraphrase2>` paraphrased sentence 2 `</paraphrase2>`
...
`<paraphrasen>` paraphrased sentence n `</paraphrasen>`
**Input sentence:** `{sentence}` **Required answer:** `{answer}`
Please return only the paraphrases in the specified format.

---

**Paraphrasing – Long-form QA**

You are given an input question sentence. Your task is to generate n diverse paraphrases of this question. You can paraphrase by using synonyms, changing sentence structure, or rephrasing in any other way, but each paraphrase should preserve the original question meaning and lead to similar answers.
Format your output as follows:
`<paraphrase1>` paraphrased sentence 1 `</paraphrase1>`
`<paraphrase2>` paraphrased sentence 2 `</paraphrase2>`
...
`<paraphrasen>` paraphrased sentence n `</paraphrasen>`
**Input question:** `{sentence}`
Please return only the paraphrases in the specified format.

---

Table 6: **Accuracy across datasets and query variants (LLaMA 3.1 8B).** We report accuracy for original queries, synthetically generated paraphrased queries, and paraphrased queries with fixed retrieval. Across all settings, accuracy remains relatively similar, indicating that paraphrasing and retrieval shifts have limited effect on final answer correctness on average. See result for `Qwen-2.5-3B` model in Table 8.

| **Short-form & Multi-hop QA: Accuracy (%)** | | | | | | | | | |
| --- | --- | --- | --- | --- | --- | --- | --- | --- | --- |
| **Dataset** | **Original Queries** | | | **Paraphrased Queries** | | | **Paraphrased (Fixed Docs)** | | |
| | EM | F1 | RM | EM | F1 | RM | EM | F1 | RM |
| `TriviaQA` | 56.0 | 66.1 | 74.0 | 55.0 | 64.4 | 73.3 | 58.7 | 67.3 | 75.0 |
| HotpotQA | 37.0 | 44.1 | 42.0 | 36.4 | 43.5 | 42.4 | 33.7 | 40.7 | 39.4 |
| 2Wiki | 28.0 | 33.9 | 37.0 | 25.9 | 31.3 | 32.7 | 26.9 | 31.7 | 33.3 |
| MuSiQue | 8.0 | 15.3 | 12.0 | 8.3 | 14.1 | 11.0 | 11.0 | 17.5 | 15.0 |

| **Long-form QA: Accuracy (%)** | | | | | | |
| --- | --- | --- | --- | --- | --- | --- |
| **Dataset** | **Original Queries** | | **Paraphrased Queries** | | **Paraphrased (Fixed Docs)** | |
| | ROUGE | LLM-Acc | ROUGE | LLM-Acc | ROUGE | LLM-Acc |
| `ELI5` | 21.9 | 74.0 | 20.7 | 71.3 | 20.8 | 70.3 |

## A.3 LLM-JUDGE CONSISTENCY EVALUATION

To assess semantic consistency between generated outputs across paraphrased queries, we employ LLM-based pairwise judgments as part of our evaluation pipeline. These judgments are designed to determine whether two answers convey the same core information, even if they differ in surface form. For all `LLaMA 3.1 8B` experiments, we use `LLaMA 3.3 70B` as the evaluator. For all `Qwen 2.5 3B` experiments, we use `GPT-4o`, an external closed-source model, as the evaluator. An example evaluation prompt is shown below:

---

**Consistency Evaluation – Long-form QA**

You are an evaluator assessing two different answers that were generated for the same question. Your task is to determine if the two answers are consistent with each other.
Consider them consistent if they present the same core information about the underlying question.
Consider them inconsistent if they provide different information, present different facts, or address the underlying question in different ways.
Based on this, reply with only `yes` or `no`. Do not output anything else.
**Answer 1:** `{output_i}`
**Answer 2:** `{output_j}`
**Are these two answers consistent? (yes/no). Response:**

---

**Consistency Evaluation – Short-form QA**

Given the following two outputs sentences, reply with `yes` if the two sentences are consistent with each other, or `no` if they are not. Do not output anything else.
**Sentence 1:** `{output_i}`
**Sentence 2:** `{output_j}`
**Are these sentences consistent? (yes/no). Response:**

---

Table 7: **Disentangling sources of inconsistency in RAG systems (`Qwen-2.5-3B`)**. Retriever consistency is low across datasets, suggesting that paraphrased queries often retrieve non-overlapping documents. This introduces context variability that is reflected in the end-to-end consistency scores. Fixing retrieval improves consistency, but variation remains, revealing the generator's sensitivity to input phrasing even with identical evidence.

| Dataset | End-to-End Consistency | | Generator (LLM) Consistency | | Retriever Consistency |
|---------|---------|-----------|---------|-----------|-----------------|
|         | Lexical | LLM-Judge | Lexical | LLM-Judge | Jaccard Overlap |
| TriviaQA | 47.9 | 73.0 | 58.6 | 87.5 | 32.5 |
| HotpotQA | 32.7 | 63.6 | 48.0 | 77.3 | 46.0 |
| 2Wiki    | 32.3 | 62.6 | 44.6 | 70.7 | 52.4 |
| MuSiQue  | 25.7 | 49.5 | 45.7 | 67.3 | 36.6 |
| Eli5     | 6.6  | 35.3 | 14.4 | 62.3 | 27.1 |

Table 8: **Accuracy across datasets and query variants (`Qwen-2.5-3B`).** We report accuracy for original queries, synthetically generated paraphrased queries, and paraphrased queries with fixed retrieval. Across all settings, accuracy remains relatively similar, indicating that paraphrasing and retrieval shifts have limited effect on final answer correctness on average.

| Short-form & Multi-hop QA: Accuracy (%) | | | | | | | | | |
|---------|------|------|------|------|------|------|------|------|------|
| Dataset | Original Queries | | | Paraphrased Queries | | | Paraphrased (Fixed Docs) | | |
|         | EM   | F1   | RM   | EM   | F1   | RM   | EM   | F1   | RM   |
| TriviaQA | 42.0 | 50.7 | 58.0 | 46.3 | 54.1 | 64.3 | 43.0 | 51.1 | 62.3 |
| HotpotQA | 20.0 | 28.3 | 37.0 | 20.9 | 27.7 | 38.4 | 18.2 | 26.4 | 38.4 |
| 2Wiki    | 13.0 | 20.4 | 36.0 | 11.1 | 19.8 | 34.7 | 12.5 | 20.2 | 32.3 |
| MuSiQue  | 4.0  | 10.0 | 9.0  | 6.0  | 9.9  | 8.0  | 5.3  | 9.7  | 7.7  |

| Long-form QA: Accuracy (%) | | | | | | |
|---------|-------|---------|-------|---------|-------|---------|
| Dataset | Original Queries | | Paraphrased Queries | | Paraphrased (Fixed Docs) | |
|         | ROUGE | LLM-Acc | ROUGE | LLM-Acc | ROUGE | LLM-Acc |
| ELI5    | 22.1  | 38.0    | 21.5  | 37.3    | 20.8  | 35.7    |

Table 9: **Comparison between Con-RAG vs. Baselines (Short-form QA Tasks) (`Qwen-2.5-3B`).** Lexical consistency measured via BLEU score while and information consistency measured using an LLM-judge. Con-RAG is trained with a group-similarity reward plus an accuracy reward (no KL), and consistently yields higher end-to-end and generator-only consistency while also improving accuracy over original queries.

| Dataset | Method | Accuracy (%) | | | End-to-End Consistency (%) | | Generator (LLM) Consistency (%) | |
|---|---|---|---|---|---|---|---|---|
| | | EM | F1 | RM | Lexical | Inform. | Lexical | Inform. |
| TriviaQA | RAG | 42.0 | 50.7 | 58.0 | 47.9 | 73.0 | 58.6 | 87.5 |
| | DRAG | 42.0 | 50.7 | 58.0 | 47.9 | 73.5 | 58.6 | 84.7 |
| | CoT-RAG | 37.0 | 44.5 | 61.0 | 41.1 | 72.3 | 52.2 | 82.3 |
| | SFT | 35.0 | 40.4 | 43.0 | 53.3 | 72.2 | 73.4 | 85.0 |
| | **Con-RAG** | **60.0** | **66.0** | **68.0** | **67.1** | **81.8** | **80.5** | **89.5** |
| HotpotQA | RAG | 20.0 | 28.3 | 37.0 | 32.7 | 63.6 | 48.0 | 77.3 |
| | DRAG | 20.0 | 28.3 | 37.0 | 32.7 | 64.3 | 48.0 | 76.8 |
| | CoT-RAG | 29.0 | 32.8 | 37.0 | 28.5 | 63.6 | 35.7 | 71.2 |
| | SFT | 30.0 | 35.4 | 32.0 | 63.3 | 77.1 | 74.5 | 85.7 |
| | **Con-RAG** | **36.0** | **43.1** | **38.0** | **64.6** | **78.2** | **77.8** | **86.7** |
| MuSiQue | RAG | 4.0 | 10.0 | 9.0 | 25.7 | 49.5 | 45.7 | 67.3 |
| | DRAG | 4.0 | 10.0 | 9.0 | 25.7 | 50.3 | 45.7 | 69.2 |
| | CoT-RAG | 5.0 | 10.9 | 9.0 | 18.1 | 52.0 | 26.5 | 62.0 |
| | SFT | 25.0 | 30.6 | 27.0 | 57.7 | 65.3 | 69.8 | 77.2 |
| | **Con-RAG** | **27.0** | **31.9** | **28.1** | **69.8** | **70.1** | **70.4** | **82.0** |
| 2Wiki | RAG | 13.0 | 20.4 | 36.0 | 32.3 | 62.6 | 44.6 | 70.7 |
| | DRAG | 13.0 | 20.4 | 36.0 | 32.3 | 63.0 | 44.6 | 70.9 |
| | CoT-RAG | 23.0 | 27.0 | 30.0 | 23.5 | 62.3 | 32.7 | 67.0 |
| | SFT | **37.0** | **38.9** | **38.0** | **70.9** | 75.8 | **84.8** | 86.9 |
| | **Con-RAG** | **37.0** | 38.4 | 37.0 | 68.2 | **76.6** | **84.8** | **89.1** |

Table 10: **Comparison between Con-RAG vs. Baselines (Long-form QA Task).** Con-RAG is trained using only the group-similarity reward with a small KL regularizer (no accuracy supervision). Despite no ground-truth, it achieves the best end-to-end and generator consistency and also improves answer quality over baselines, whereas SFT on reference answers underperforms in this open-ended setting (`Qwen 2.5 3B`).

| Dataset | Method | Accuracy (%) | | End-to-End Consistency (%) | | Generator (LLM) Consistency (%) | |
|---|---|---|---|---|---|---|---|
| | | ROUGE | LLM-Acc | Lexical | Inform. | Lexical | Inform. |
| `ELI5` | RAG | 22.1 | 38.0 | 6.6 | 35.3 | 14.4 | 62.3 |
| | DRAG | 22.1 | 38.0 | 6.6 | 35.3 | 14.4 | 63.8 |
| | CoT-RAG | 21.1 | 36.0 | 4.9 | 34.0 | 9.6 | 55.5 |
| | SFT | **24.3** | 36.0 | 5.4 | 17.2 | 7.0 | 19.0 |
| | **Con-RAG** | 22.6 | **58.0** | **9.3** | **42.8** | **17.9** | **67.5** |

Table 11: **Effect of Inference Temperature on Standard RAG(`ELI5 - Qwen-2.5-3B`).** We vary only the decoding temperature $T$ at inference to study its effect on consistency and accuracy. Moderate temperature ($T = 0.5$) improves LLM agreement and lexical consistency compared to deterministic decoding ($T = 0.0$), while preserving accuracy. However, higher temperatures ($T \geq 1.0$) degrade both consistency and accuracy, with outputs at $T = 2.0$ nearly collapsing.

| $T$ | Accuracy (%) | | End-to-End Cons. (%) | | Generator Cons. (%) | |
|---|---|---|---|---|---|---|
| | ROUGE | LLM-Acc | Lexical | LLM-Judge | Lexical | LLM-Judge |
| 0.0 | 22.1 | 38.0 | 6.6 | 35.3 | 14.4 | 62.3 |
| 0.5 | **21.4** | **52.0** | **10.4** | **37.7** | **15.2** | **65.3** |
| 1.0 | 21.8 | 48.0 | 2.5 | 34.0 | 5.2 | 59.5 |
| 2.0 | 6.1 | 0.0 | 0.1 | 2.0 | 0.2 | 1.5 |

