# OpenReview forum: "Improving Consistency in Retrieval Augmented Systems with Group Similarity Reward"
_ICLR.cc/2026/Conference — Submitted to ICLR 2026_

### Official Review · Reviewer_UNQA · 2025-10-25

**Soundness:** 3
**Presentation:** 3
**Contribution:** 2
**Rating:** 4
**Confidence:** 3

**Summary:**

The paper addresses the issue of information inconsistency in Retrieval-Augmented Generation (RAG) systems, where semantically equivalent queries (paraphrases) often yield inconsistent answers due to variations in both retrieved evidence and generator sensitivity. To tackle this, the authors define and measure consistency at three levels: retriever, generator, and end-to-end. They propose PS-GRPO (Paraphrased Set Group Relative Policy Optimization), a reinforcement learning method that encourages consistent outputs across paraphrases of the same query using a group similarity reward, with scalable sampling-based approximations. The resulting model, Con-RAG, is evaluated on short, multi-hop, and long-form QA datasets (TriviaQA, HotpotQA, 2WikiMultiHopQA, MuSiQue, ELI5), showing significant improvements in both generator- and end-to-end consistency while maintaining or improving factual accuracy. Even without explicit ground-truth supervision, Con-RAG enhances factual alignment as judged by LLM-based evaluators, highlighting its robustness for reliable RAG applications.

**Strengths:**

• I like that the paper disentangles the contributions of the retriever, generator, and overall pipeline to consistency — it really helps clarify where inconsistency comes from. The analysis also shows that paraphrasing and retrieval shifts have only a limited effect on final answer correctness on average, which is an important and somewhat reassuring finding.

• A particularly interesting result is on long-form QA (ELI5): even without explicit ground-truth supervision, Con-RAG significantly improves both end-to-end consistency and LLM-judged factual accuracy. That’s quite impressive and suggests the optimization signal generalizes well beyond short-answer settings.

• I appreciate that the paper treats consistency as more than just lexical overlap. By centering on information consistency and combining lexical and LLM-judge metrics, the authors offer a much more robust perspective. The fact that optimizing for consistency can jointly improve factuality (as reflected in LLM-Acc) is a strong and nuanced contribution.

**Weaknesses:**

• The experimental setup feels somewhat limited in terms of model diversity and scale. It’s unclear whether the observed gains would hold — or even become more pronounced — with larger or more capable base models. Including results on stronger LLMs would make the conclusions more convincing.

• I’m not fully convinced about how novel PS-GRPO is. The main innovation seems to lie in computing the n×g similarity matrix, but conceptually this could be simplified to a scalar 0/1 reward—treating paraphrases closer to the ground truth as 1 and others as 0—then applying standard GRPO training. It would be helpful to clarify what new learning dynamics PS-GRPO introduces beyond this simplification.

**Questions:**

• The paper shows that even without explicit ground-truth supervision, the method improves end-to-end consistency and LLM-judged factual accuracy, outperforming SFT — but it’s not entirely clear whether this effect is specific to ELI5 or generalizable to other long-form QA tasks. Can the same conclusion extend to short-form or multi-hop settings as well? Some discussion or evidence on generalizability would strengthen the claim.

• While the paper disentangles the retriever and generator effects, it still remains unclear how retriever shifts and LLM sensitivity concretely influence end-to-end consistency. What are their relative contributions? Moreover, the relationship between this sensitivity and final factual accuracy is not clearly analyzed.

• From Table 4, it seems that higher consistency correlates with better accuracy, but it’s unclear how incorrect consistency (i.e., consistently wrong answers) compares to correct consistency. Clarifying this distinction would help interpret whether consistency itself is always desirable or potentially misleading.

---

> ### Author Response · Authors · 2025-12-04
>
> We thank the reviewer for their review!
>
>
> **On model diversity and scale.**
>
> We restrict our experiments to 7–8B parameter models due to computational resources available. We now include additional results on the Natural Questions dataset (Table C), where we continue to  observe consistent improvements from Con-RAG. Our experiments span two strong open-source model families LLaMA-3.1-8B and Qwen-2.5-3B, , both commonly used research models that offer strong performance. Our evaluation spans six QA datasets covering short-form, multi-hop, and long-form reasoning (TriviaQA, HotpotQA, 2Wiki, MuSiQue, ELI5, NQ), includes five strong baselines (RAG, DRAG, CoT-RAG, SFT, Standard GPRO), and uses both lexical and LLM-judge metrics for accuracy and consistency. This setup provides a solid foundation for assessing the generality of Con-RAG.
>
> **On the novelty of PS-GRPO**
> Our approach does not require any ground-truth supervision to compute rewards. This is a central property of PS-GRPO: the model improves information consistency purely from cross-paraphrase similarity, without using reference answers. This makes the method applicable to long-form and open-ended QA settings where single gold answers are unavailable or insufficient.
>
> We have added a new baseline where we run standard GRPO and give each completion a scalar reward equal to its BLEU score with the ground-truth answer (instead of using the n \times g group similarity matrix). This corresponds exactly to the “simplified” scheme the reviewer describes: a pointwise supervised reward against reference answers, followed by standard GRPO training. Empirically, PS-GRPO outperforms this ground-truth GRPO variant on both consistency and accuracy across QA tasks (see new experiment in Table C). This suggests that: The learning signal from mutual agreement across paraphrases is more effective than simply pushing each response toward a single reference.
>
> **Table C: Natural Questions Dataset (NQ) Results**
>
> | Dataset | Method   | ROUGE | LLM-Acc | End-to-End Consistency (Lexical) | End-to-End Consistency (Inform.) | Generator Consistency (Lexical) | Generator Consistency (Inform.) |
> |---------|----------|--------|---------|----------------------------------|----------------------------------|----------------------------------|----------------------------------|
> | NQ      | RAG      | 0.269  | 0.61    | 0.142                            | 0.532                            | 0.224                            | 0.682                            |
> |      | DRAG     | 0.269  | 0.59    | 0.142                            | 0.535                            | 0.224                            | 0.682                            |
> |      | CoT-RAG  | 0.243  | 0.55    | 0.121                            | 0.547                            | 0.198                            | 0.705                            |
> |      | SFT      | 0.368  | 0.22    | 0.371                            | 0.362                            | 0.562                            | 0.550                            |
> |      | GRPO     | 0.374  | 0.52    | 0.192                            | 0.475                            | 0.296                            | 0.605                            |
> |      | Con-RAG  | 0.235  | 0.59    | 0.228                            | 0.593                            | 0.323                            | 0.755                            |
>
> ---
>
> **Generalization beyond ELI5 Dataset**
>
> Thank you for the question about generalizability. Our method improves consistency and factual accuracy, outperforming SFT  even without explicit ground-truth supervision is demonstrated on ELI5, where the open-ended nature of answers makes label-free training natural. That said, we see similar behavior beyond ELI5. In particular, Table 5 (“Effect of Accuracy Reward Variant on Con-RAG (TriviaQA – Qwen-2.5-3B)”) includes a consistency-only ablation (using only group similarity rewards with no accuracy term), and this variant already achieves strong accuracy and substantial consistency gains over the baseline approaches. Together with our new additional consistency-only experiment on Natural Questions (NQ), where Con-RAG trained without any ground-truth reward also improves both EM/F1 and consistency over RAG (see Table C).

---

> > ### Author Response · Authors · 2025-12-04
> >
> > **Disentangles the retriever and generator effects**
> >
> > We estimate the generator’s paraphrase sensitivity by freezing the retrieved evidence across all paraphrases of a canonical query: for a given ($q_0$), we fix ($R(q_0)$) and run the LLM on each paraphrase ($p_i$) with the same document set, measuring pairwise similarity over (${y_i^{\text{fixed}}}$). This gives the “Generator Consistency” columns in Table 1. In contrast, “End-to-End Consistency” allows both retrieval and generation to vary (($R(p_i)$) and ($y_i = \text{LLM}(p_i, R(p_i)))$). The retriever’s contribution is captured in two ways: (i) directly, via the Jaccard overlap of retrieved sets across paraphrases (“Retriever Consistency”), and (ii) indirectly, via the gap between end-to-end and generator-only consistency. Empirically, generator consistency is consistently higher than end-to-end consistency across all datasets (e.g., on TriviaQA: 67.3 vs. 53.0 lexical, 88.5 vs. 77.8 information-level), while Jaccard overlaps remain modest (e.g., 27–52). This indicates that both components matter, but retrieval shifts are a major driver of the drop from generator-only to end-to-end consistency: once the evidence is fixed, a substantial fraction of the inconsistency disappears, yet a non-trivial portion remains due to the LLM’s sensitivity to paraphrasing.
> >
> > ---
> >
> > **Consistency Correlation with Accuracy**
> >
> > Our goal is indeed not to optimize “consistency at all costs,” but to move models toward being both consistent and correct. First, Table 4 always reports consistency together with accuracy on the same benchmark. If Con-RAG were mainly making the outputs “consistent and wrong,” we would expect accuracy on the original (canonical) queries to drop, which we do not observe—instead, both accuracy and consistency improve.
> >
> > We explicitly acknowledge that our method can, in some cases, lead to consistent but wrong  outputs on certain inputs. However, even in such cases, consistency is still preferable to being both inconsistent and wrong, the latter produces more unpredictable and contradictory outputs. Importantly, in all our experiments we do not observe this trade-off dominating in practice: consistency improvements are accompanied by equal or higher accuracy. Future work could more deeply characterize when consistency and accuracy align or diverge, especially in long-form, open-ended settings.

---

### Official Review · Reviewer_4xia · 2025-10-26

**Soundness:** 2
**Presentation:** 2
**Contribution:** 2
**Rating:** 4
**Confidence:** 2

**Summary:**

Authors study information consistency in RAG, decomposing it into retriever, generator, and end-to-end components, and propose Con-RAG trained with Paraphrased-Set GRPO (PS-GRPO): sample multiple rollouts across paraphrases of a canonical query and reward outputs that are similar within the paraphrase group. They also give a scalable reward approximation (subsampling paraphrases/rollouts) to keep training tractable.

**Strengths:**

1. PS-GRPO is clearly specified: group-normalized advantages, clipped objective, and an explicit group-similarity reward over paraphrases, with optional accuracy reward when ground truth exists.

2. A useful consistency framework that separates retriever Jaccard, generator consistency under fixed evidence, and end-to-end behavior; this makes ablations actionable.

3. Sensible evaluation choices like deterministic decoding to remove sampling noise and a reward-metric ablation that compares BLEU-n, ROUGE-L, EM as similarity functions.

**Weaknesses:**

1. Paraphrase generation includes the gold answer in the prompt, which risks target leakage and may inflate consistency by narrowing the paraphrase space.

2. The core reward is lexical (BLEU); optimizing surface overlap is not the same as optimizing information-level agreement

3. The retriever remains the principal bottleneck, yet the method fine-tunes only the generator; no retriever co-training or simple stabilizers are evaluated.

**Questions:**

1. Section A.2 (L108–121): if paraphrases are generated without gold answers, how much do consistency and accuracy drop? Please quantify across tasks and share examples where answer-priming mattered.

2. Section 2.2 (L25–41): how sensitive are gains to κ and s in the subsampled reward—what’s the best compute-quality frontier for practitioners with fixed budgets?

3. Section 2.1 (L182–191): when retriever Jaccard is particularly low, do generator-only gains still lift end-to-end consistency, or do they saturate? Any observed failure modes?

---

> ### Author Response · Authors · 2025-12-04
>
> We thank the reviewer for their review!
>
> ---
>
> **Regarding gold answer while generating paraphrases**
>
> We acknowledge the concern and clarify why the gold answer is included. In early experiments, paraphrases generated without the gold answer degraded the quality of paraphrases generated: they remained similar to the original query but  slightly changed the implied answer or introduced new answerable variants. This directly harmed the validity of our consistency evaluation. Including the gold answer anchored the paraphrase generation to the same underlying fact. For example, given the original question “What nationality was Person X?” (answer: American), a paraphrase generated without the gold answer was “What was the ethnic background of Person X?”.  Providing the gold answer resolved such issues, and in practice we observed no cases where the gold answer leaked into the paraphrases.
>
> To further demonstrate this, we now generate paraphrases on HotpotQA both with and without ground truth and compare accuracy across original queries and paraphrased queries. As shown in Table B below, with the gold answer included, paraphrased queries maintain nearly identical accuracy to the original queries (e.g., EM drops from 37.0 → 36.4). In contrast, when the gold answer is omitted, accuracy on paraphrased queries drops sharply (EM: 37.0 → 27.4), indicating that many paraphrases drift semantically. Importantly, this does not affect the model being evaluated. Paraphrases are generated offline, and the ground truth is never shown to the RAG model.
>
> We note that for long-form QA tasks  (e.g., ELI5, NQ) we do not include ground truth during paraphrase generation, and this issue does not arise (see prompts in Appendix A2). We will include these discussions and results in the revised paper.
>
>  **Table B: Accuracy over paraphrases generated with and without ground truth**
>
> | Setting               | Query Type  | EM   | F1   | RM   |
> |----------------------|-------------|------|------|------|
> | —                    | Original    | 37.0 | 44.1 | 42.0 |
> | **With Gold Answer** | Paraphrased | 36.4 | 43.5 | 42.4 |
> | **Without Gold Answer** | Paraphrased | 27.4 | 36.1 | 35.4 |
>
> ---
>
> **BLEU metric as a proxy signal**
>
> Thank you for raising this. We fully agree that BLEU metrics alone cannot fully capture semantic consistency. In our formulation the BLEU metric serves as a lightweight proxy signal that nudges the model towards more consistent outputs across paraphrases. Our use of BLEU is driven by scalability considerations; group-level reward computation must be performed hundreds of times per optimization step, making an efficient metric essential. Moreover, prior work [2] (and our own ablations in Table 4) shows that lower-order BLEU correlates surprisingly well with human preference signals.
>
> Importantly, even under this minimal reward, Con-RAG improves not only lexical overlap but also LLM-judge consistency, which directly evaluates information-level agreement. While stronger semantic rewards such as an LLM-based consistency critic could be used, they are prohibitively expensive at the scale. Our goal in this paper is to demonstrate that (1) paraphrased-set RL is an effective framework for optimizing consistency, and (2) even with simple and efficient similarity rewards, the method produces measurable gains in information consistency and factual accuracy.
>
> In addition, we emphasize that the improvements are not simply a consequence of reduced output diversity. By tying multiple paraphrased inputs of the same query together during training, Our approach effectively acts as some form of regularization: the training objective pushes the model to capture the invariant semantic structure across paraphrases instead of lexical differences. This operates similarly to data augmentation, where multiple rephrasings expose the model to diverse realizations of the same underlying input. As a result, the model becomes more robust to paraphrasing. This is reflected in our results: Con-RAG not only yields higher consistency scores but also improves factual accuracy (e.g., highest LLM-judge accuracy and ROUGE in Table 3)  even in the absence of explicit ground-truth supervision.
>
> [2] Yapei et al., Bleuberi: Bleu is a surprisingly effective reward for instruction following.

---

> ### Author Response · Authors · 2025-12-04
>
> **Retrieval Bottleneck**
>
> Our method is designed to be retriever-agnostic and can be applied on top of any retrieval system, robust or not, without requiring changes to the retrieval pipeline. Moreover, our analysis shows that even when the retriever is completely robust, i.e., when we freeze the retrieved documents across paraphrases, substantial inconsistency remains at the generator level (Generator Consistency; see Table 2 & 3). This motivates our emphasis on improving the generator to remain robust to retrieval-induced variability. We also note that our approach is not limited to RAG: it applies broadly to standalone general LLMs, where paraphrase sensitivity and generation inconsistency persist even without retrieval. We focus on RAG because the variability introduced by the retriever amplifies these inconsistencies and makes the problem more critical in practice. Extending our framework to jointly co-train or regularize both the retriever and generator is an interesting direction for future work.
>
> ---
>
> **Sensitivity to κ and s in subsampled reward**
>
> To assess the computational trade-offs introduced by the relaxed group-similarity reward, we performed a controlled sweep over the subsampling parameters $ \kappa$  (number of paraphrases sampled) and  $s$  (rollouts per paraphrase) on the NQ (Natural Questions) dataset while logging CPU/GPU energy and power using the CodeCarbon python package [3]. Results are summarized in Table A below. We find that Con-RAG’s performance is stable across a wide range of ((\kappa), (s)) settings: increasing $\kappa$ and  $s$ yields only mild fluctuations in accuracy and consistency metrics. However, the computational footprint (GPU and CPU Power) grows noticeably with larger subsampling budgets. These results highlight that the reward signal is not highly sensitive to these hyperparameters, meaning practitioners can safely adopt low-cost settings which already provide strong consistency gains at a fraction of the compute. For high-resource settings, larger ((\kappa, s)) provide marginal additional improvements but at a noticeably higher cost, defining a compute-quality frontier for deployment scenarios with fixed budgets.
>
> ---
>  **Table A: Natural Questions Dataset Results Across (κ, s) Settings**
>
> | Dataset | κ | s | ROUGE | LLM-Acc | End-to-End Consistency (Lexical) | End-to-End Consistency (Inform.) | Generator Consistency (Lexical) | Generator Consistency (Inform.) | CPU Energy (W) | GPU Power (W) |
> | ------- | - | - | ----- | ------- | -------------------------------- | -------------------------------- | ------------------------------- | ------------------------------- | -------------- | ------------- |
> | NQ | 1 | 1 | 0.252 | 0.600 | 0.213 | 0.605 | 0.307 | 0.708 | 118 | 441 |
> | | 2 | 1 | 0.241 | 0.590 | 0.313 | 0.593 | 0.427 | 0.740 | 120 | 457 |
> | | 5 | 1 | 0.243 | 0.600 | 0.284 | 0.593 | 0.385 | 0.730 | 123 | 462 |
> | | 5 | 5 | 0.256 | 0.630 | 0.235 | 0.598 | 0.339 | 0.762 | 135 | 492 |
>
>
> ---

---

### Official Review · Reviewer_eGuv · 2025-10-30

**Soundness:** 2
**Presentation:** 3
**Contribution:** 2
**Rating:** 4
**Confidence:** 4

**Summary:**

This paper addresses the issue of inconsistent outputs in RAG systems when handling semantically equivalent queries. It proposes PS-GRPO, a reinforcement learning method that uses group similarity rewards across paraphrased query sets to encourage consistent generation. The approach is applied to build Con-RAG, a consistency-optimized RAG model.

**Strengths:**

1.	The proposed relaxed group similarity reward is interesting and effectively reduces computational cost.
2.	The decomposition into retriever-level, generator-level, and end-to-end consistency is elegant and interpretable.

**Weaknesses:**

1.	There is no detailed analysis provided on why Con-RAG outperforms SFT in short-form QA tasks (Table 2), leaving the underlying factors behind the performance improvement insufficiently explained.
2.	The paper does not explicitly address how to distinguish between diverse-but-correct and inconsistent answers in open-ended QA tasks such as ELI5. The proposed group similarity reward optimizes for surface-level similarity (e.g., BLEU) and lacks mechanisms to capture semantic equivalence, making it unclear whether the observed “consistency” reflects genuine informational alignment or simply reduced diversity. Similarly, the evaluation metrics—largely based on lexical overlap and sparse LLM judgments—are insufficient to rigorously assess semantic consistency in tasks with multiple valid answers. These limitations should be explicitly discussed and empirically analyzed.
3.	The paper does not report comparative information on training time and GPU resources.

**Questions:**

My questions are listed in "Weaknesses". I am looking forward to clear explanations from the authors.
I will revise my rating according to the author's feedback and the reviewer's discussion.

---

> ### Author Response · Authors · 2025-12-04
>
> We thank the reviewer for their review!
>
> ---
>
> **Why Con-RAG outperforms SFT for Short-Form QA**
>
> Thank you for highlighting the need for a deeper discussion. SFT treats each paraphrase-answer pair independently and minimizes token-level cross-entropy with a single reference answer (or ground truth). This encourages the model to reproduce the surface form of that reference and penalizes any deviation (even when multiple phrasings are equally correct). For instance, a date like “November 16th, 2025” may be expressed accurately in several valid forms (“Nov. 16, 2025,” “16 November 2025,” etc.), yet SFT penalizes all but the exact reference. This also explains why the F1-based accuracy reward was most effective in our experiments compared to Exact match (EM) and Relaxed Match (RM) as it provides a more flexible alignment signal that tolerates variations in phrasing (see Table 5). In contrast, Con-RAG explicitly optimizes a group-level objective over paraphrase sets. Outputs are rewarded not only for matching the ground truth but also for being consistent with outputs generated for other paraphrases of the same underlying question. This means that alternative answer formulations can still receive positive reward if they remain consistent across paraphrases, rather than being penalized for surface-level differences.
>
> One could however have a variant of our approach for SFT which has a loss term that penalizes inconsistency across paraphrases. However, recent evidence further supports our design choice of using an RL approach rather than SFT to optimize consistency: SFT Memorizes, RL Generalizes [1] shows that SFT largely encourages memorization of surface patterns, causing models to overfit to the exact phrasing and structure of training examples, while RL, with outcome-level rewards, learns transferable rules that robustly generalize to unseen variants. Thus, Con-RAG’s gains over SFT align with the broader finding that RL is inherently better suited for learning structures that must remain stable across input variations. We will incorporate this expanded discussion into the revised paper.
>
> [1] Chu, Tianzhe, et al. "Sft memorizes, rl generalizes: A comparative study of foundation model post-training."
>
> ---
>
> **BLEU metric as a proxy signal**
>
> Thank you for raising this. We fully agree that BLEU metrics alone cannot fully capture semantic consistency. In our formulation the BLEU metric serves as a lightweight proxy signal that nudges the model towards more consistent outputs across paraphrases. Our use of BLEU is driven by scalability considerations; group-level reward computation must be performed hundreds of times per optimization step, making an efficient metric essential. Moreover, prior work [2] (and our own ablations in Table 4) shows that lower-order BLEU correlates surprisingly well with human preference signals.
>
> Importantly, even under this minimal reward, Con-RAG improves not only lexical overlap but also LLM-judge consistency, which directly evaluates information-level agreement. While stronger semantic rewards such as an LLM-based consistency critic could be used, they are prohibitively expensive at the scale. Our goal in this paper is to demonstrate that (1) paraphrased-set RL is an effective framework for optimizing consistency, and (2) even with simple and efficient similarity rewards, the method produces measurable gains in information consistency and factual accuracy.
>
> In addition, we emphasize that the improvements are not simply a consequence of reduced output diversity. By tying multiple paraphrased inputs of the same query together during training, Our approach effectively acts as some form of regularization: the training objective pushes the model to capture the invariant semantic structure across paraphrases instead of lexical differences. This operates similarly to data augmentation, where multiple rephrasings expose the model to diverse realizations of the same underlying input. As a result, the model becomes more robust to paraphrasing. This is reflected in our results: Con-RAG not only yields higher consistency scores but also improves factual accuracy (e.g., highest LLM-judge accuracy and ROUGE in Table 3)  even in the absence of explicit ground-truth supervision.
>
> [2] Yapei et al., Bleuberi: Bleu is a surprisingly effective reward for instruction following.

---

> ### Author Response · Authors · 2025-12-04
>
> **Training-Time / Resource Comparison**
>
> To assess the computational trade-offs introduced by the relaxed group-similarity reward, we performed a controlled sweep over the subsampling parameters $\kappa$  (number of paraphrases sampled) and  $s$  (rollouts per paraphrase) on the NQ (Natural Questions) dataset while logging CPU/GPU energy and power using the CodeCarbon python package [3]. Results are summarized in Table A below. We find that Con-RAG’s performance is stable across a wide range of (($\kappa$), (s)) settings: increasing $\kappa$ and  $s$ yields only mild fluctuations in accuracy and consistency metrics. However, the computational footprint (GPU and CPU Power) grows noticeably with larger subsampling budgets. These results highlight that the reward signal is not highly sensitive to these hyperparameters, meaning practitioners can safely adopt low-cost settings which already provide strong consistency gains at a fraction of the compute. For high-resource settings, larger (($\kappa$, $s$)) provide marginal additional improvements but at a noticeably higher cost, defining a compute-quality frontier for deployment scenarios with fixed budgets.
>
> ---
>  **Table A: Natural Questions Dataset Results Across (κ, s) Settings**
>
> | Dataset | κ | s | ROUGE | LLM-Acc | End-to-End Consistency (Lexical) | End-to-End Consistency (Inform.) | Generator Consistency (Lexical) | Generator Consistency (Inform.) | CPU Energy (W) | GPU Power (W) |
> | ------- | - | - | ----- | ------- | -------------------------------- | -------------------------------- | ------------------------------- | ------------------------------- | -------------- | ------------- |
> | NQ | 1 | 1 | 0.252 | 0.600 | 0.213 | 0.605 | 0.307 | 0.708 | 118 | 441 |
> | | 2 | 1 | 0.241 | 0.590 | 0.313 | 0.593 | 0.427 | 0.740 | 120 | 457 |
> | | 5 | 1 | 0.243 | 0.600 | 0.284 | 0.593 | 0.385 | 0.730 | 123 | 462 |
> | | 5 | 5 | 0.256 | 0.630 | 0.235 | 0.598 | 0.339 | 0.762 | 135 | 492 |
>
>
> ---

---

### Meta-Review · Area_Chair_3wfG · 2025-12-03

**Summary:**

This paper addresses output inconsistency in RAG systems and proposes the PS-GRPO reinforcement learning method that leverages paraphrased query sets to encourage consistent generation. The resulting model, Con-RAG, shows improvements across several QA benchmarks.

Reviewers generally appreciated:

* A clear decomposition of consistency sources into retriever, generator, and end-to-end components.
* The idea of group similarity reward and paraphrase-based training is conceptually meaningful.
* Experiments include multiple QA tasks and show improvements in consistency.

However, substantial and fundamental weaknesses dominate:
* The evaluation metrics for “consistency” are mainly lexical (BLEU/ROUGE), which does not capture semantic equivalence — raising the possibility that the model only learns surface alignment rather than informational consistency.
* It remains unclear whether improvements reflect true semantic alignment or simply reduced diversity.
* Paraphrase generation includes the gold answer in the prompt, creating strong target leakage and artificially narrowing the paraphrase space.
* Retriever–generator interaction is insufficiently studied, and the retriever remains untrained, despite being a primary source of inconsistency.
* The paper does not analyze or quantify cases of “consistent but wrong answers,” i.e., harmful consistency.
* There is no analysis of computational requirements or scalability of PS-GRPO, despite being RL-based and requiring multiple rollouts.
* Novelty of PS-GRPO relative to simpler reward structures is questionable, and authors did not provide justification or theoretical argument for why their design is necessary.

All reviewers unanimously decided to reject this paper, and the authors did not provide a rebuttal, the paper is not ready for acceptance.

**Reviewer Concerns:**

No, the authors did not provide a rebuttal.

**Reviewer Scores:**

No, the authors did not provide a rebuttal.

---

### Decision · Program_Chairs · 2026-01-26

Reject